# Orthogonal interlayer coupling in an all-antiferromagnetic junction

Yongjian Zhou[1,3], Liyang Liao[1,3], Tingwen Guo[1], Hua Bai[1], Mingkun Zhao[2], Caihua Wan[2], Lin Huang[1], Lei Han[1], Leilei Qiao[1], Yunfeng You[1], Chong Chen[1], Ruyi Chen[1], Zhiyuan Zhou[1], Xiufeng Han [2], Feng Pan[1] & Cheng Song [1✉]

In conventional ferromagnet/spacer/ferromagnet sandwiches, noncollinear couplings are commonly absent because of the low coupling energy and strong magnetization. For antiferromagnets (AFM), the small net moment can embody a low coupling energy as a sizable coupling field, however, such AFM sandwich structures have been scarcely explored. Here we demonstrate orthogonal interlayer coupling at room temperature in an all-antiferromagnetic junction $Fe_2O_3/Cr_2O_3/Fe_2O_3$, where the Néel vectors in the top and bottom $Fe_2O_3$ layers are strongly orthogonally coupled and the coupling strength is significantly affected by the thickness of the antiferromagnetic $Cr_2O_3$ spacer. From the energy and symmetry analysis, the direct coupling via uniform magnetic ordering in $Cr_2O_3$ spacer in our junction is excluded. The coupling is proposed to be mediated by the non-uniform domain wall state in the spacer. The strong long-range coupling in an antiferromagnetic junction provides an unexplored approach for designing antiferromagnetic structures and makes it a promising building block for antiferromagnetic devices.

[1] Key Laboratory of Advanced Materials (MOE), School of Materials Science and Engineering, Tsinghua University, Beijing 100084, China. [2] Beijing National Laboratory for Condensed Matter Physics, Institute of Physics, University of Chinese Academy of Sciences, Chinese Academy of Sciences, Beijing 100190, China. [3] These authors contributed equally: Yongjian Zhou, Liyang Liao. ✉email: songcheng@mail.tsinghua.edu.cn

In the magnets/spacer/magnets thin film sandwich structures, the magnetic orders can serve as a boundary condition for the emergence of novel state in the spacer, and such a state brings about the coupling between magnetic orders. The most well-established example is the giant magnetoresistance system, ferromagnet/transition metal/ferromagnet[1–5], where the electron standing wave state[6] in spacer is induced by the magnetizations in two ferromagnets (FMs) and leads to the collinear interlayer coupling. Apart from metallic spacer, the interlayer coupling can exist across antiferromagnetic[7,8] and non-magnetic[9] insulators, providing more material alternatives for devices. Noncollinear coupling may also exist in FMs/spacer/FMs[10–12], which is caused by interface roughness and the oscillatory collinear coupling[13]. But such a coupling is usually overshadowed by the collinear coupling[10], which is much easier for detection than the noncollinear coupling in FMs. In addition, the large net moment of FMs results in a small effective coupling field for a given coupling energy, which hinders the detection of imperceptible interactions such as noncollinear couplings in FMs-based systems.

The interlayer coupling in another important and common magnetic materials, antiferromagnets (AFMs)[14–16], has been long-term overlooked. However, the small net moment in AFMs can embody a low coupling energy to a sizable interlayer coupling field, enabling the detection of imperceptible interactions. Moreover, collinear parallel/antiparallel arrangements of Néel vectors in antiferromagnets are usually identical in magnetoresistance measurements, so that the small noncollinear interaction can be clearly detected in an AFM/spacer/AFM junction. Hence, AFMs have unique advantages in unveiling interlayer coupling, bringing out new opportunities to discover novel condense matter phases in the spacer.

Here, we demonstrate the unprecedented orthogonal coupling of Néel vectors (Fig. 1b) between two separated antiferromagnetic $\alpha$-$Fe_2O_3$ layers in a $Fe_2O_3$/$Cr_2O_3$/$Fe_2O_3$ junction via magneto-transport measurements and x-ray magnetic linear dichroism (XMLD) spectroscopy. $\alpha$-$Fe_2O_3$ is a high-Néel-temperature antiferromagnet[17] with a weak in-plane anisotropy and concomitant low spin-flop field[18,19], as well as sizable spin Hall magnetoresistance (SMR) signals[20–24], enabling us to control and detect its Néel vector. The coupling can be mediated by non-uniform domain wall state in the $Cr_2O_3$ spacer, which is supported by our theory model. Interlayer coupling effect via direct uniform magnetic ordering is excluded by the energy and symmetry analyses. The interlayer coupling in an all-antiferromagnetic junction not only opens new avenues to fundamental research, but also provides a potential building block for antiferromagnetic devices[25–29], which have attracted increasing attention[14,30,31].

## Results

**SMR and XMLD measurements**. We first show SMR signals of a control sample $Fe_2O_3(12)$/$Pt(4)$ (units in nanometers) in Fig. 2a, where the magnetic field ($H$) and the current ($I$) are along the x-

axis and the spin polarization generated by the spin Hall effect of Pt is along the y-axis. Comparatively low resistance states at high magnetic fields reflect that the Néel vector ($n$) of $Fe_2O_3$ is perpendicular to $H$ ($I$) due to the spin-flop at high fields and deviates towards $H$ ($I$) at low fields, which is quite a characteristic for negative SMR of AFMs[20–23]. The resistance peak owing to the deviation of $n$ from the spin-flop state appears at a negative field (−0.35 T) as sweeping the field from positive to negative (black line), indicating that the Néel vector almost keeps the spin-flop state at zero-field[19]. Note that $Fe_2O_3$ with the thickness below tens of nanometers maintains easy-plane anisotropy without Morin transition[19,21,23]. Similar SMR signals are obtained in another control sample $Cr_2O_3(4.4)$/$Fe_2O_3(4)$/$Pt(4)$ (Fig. 2b), where $Fe_2O_3$ was grown on a $Cr_2O_3$ buffer to ensure a closer scenario as the top $Fe_2O_3$ in the $Fe_2O_3$/$Cr_2O_3$/$Fe_2O_3$ junction which will be discussed below. The SMR signals of the control samples are simulated and shown in Supplementary Note 2, where the hysteresis is caused by the competition between Zeeman energy and anisotropy energy. The existence of Dzyaloshinskii-Moriya interaction (DMI) in $Fe_2O_3$[32] induces canting moment and the resultant switching hysteresis behavior. The antiferromagnetic $Cr_2O_3$ buffer possesses a spin-flop field higher than 6 T[33], which does not contribute to the observed SMR signals.

Figure 2c displays a representative high-angle annular dark-field scanning transmission electron microscopy (HAADF-STEM) image of the $Fe_2O_3(12)$/$Cr_2O_3(4.4)$/$Fe_2O_3(4)$ cross-section, reflecting the epitaxial growth of the junction (Supplementary Note 3). Figure 2d presents SMR curves of the $Fe_2O_3$/$Cr_2O_3$/$Fe_2O_3$ junction, which was covered by 4 nm-thick Pt. Note that the signal disappears in the junction with Ti as the cap layer (Supplementary Note 4), which has negligible spin Hall effect, suggesting that the signal is caused by SMR. Four typical $H$ [(i) → (iv)] are denoted in the inset. The most eminent feature is that two resistance peaks emerge when sweeping $H$ from positive to negative (black line) or reverse (red line), which is different from the SMR signals of the single $Fe_2O_3$ in Fig. 2a, b. One resistance peak appears before $H = 0$, indicating the existence of coupling effect. A low resistance is obtained for $n \perp I$ ($n$ is parallel to spin polarization) at the spin-flop state. As $H$ sweeps downward, the first resistance peak (high resistance state) at a positive $H$ [$\mu_0H = +0.3$ T, (i)] reveals that $n$ deviates from the spin-flop state and is unexpectedly aligned along $n$ // $H$ ($I$). This observation indicates that another effect suppresses the magnetic field effect. We attribute the overwhelming effect to the interlayer coupling between two $Fe_2O_3$ layers through the $Cr_2O_3$ spacer. The AFM coupling generates an orthogonal (90°) arrangement of $n$ in two $Fe_2O_3$ layers. The coupling between net moment in $Cr_2O_3$ at high temperature and $n$ in $Fe_2O_3$ is excluded (Supplementary Note 5 and 6). Based on the magnetic field (Fig. 2a, b) and angle-dependent SMR measurements (Supplementary Note 7) in $Fe_2O_3$/$Pt$ and $Cr_2O_3$/$Fe_2O_3$/$Pt$ control samples, we find that the top thinner $Fe_2O_3$ possesses a lower spin-flop field than its bottom thicker counterpart, in analogy to a soft ferromagnet with small coercivity. Because of the relatively lower spin-flop field and smaller Zeeman energy of the top thinner $Fe_2O_3$, the $n$ in the top $Fe_2O_3$ has the priority to deviate from the spin-flop state as a result of the interlayer coupling, resulting in the resistance peak before zero-field. This is bolstered by the simulation results based on calculating the energy profile of different magnetic configurations in Fig. 2e (Supplementary Note 2).

As $H$ sweeps to the negative side, the SMR signal decreases and a resistance valley appears at negative $H$ (ii), which is almost the same as the location of resistance peak in Fig. 2a. This indicates that the direction of $n$ in the bottom $Fe_2O_3$ is $n$ // $H$, and the

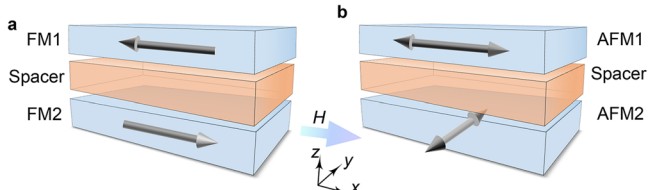

**Fig. 1 Two types of interlayer coupling in magnets. a** Illustration of the antiparallel interlayer coupling in the top and bottom ferromagnetic layers (FM1 and FM2) **b** Illustration of the orthogonal interlayer coupling between antiferromagnets (AFM1 and AFM2) found here.

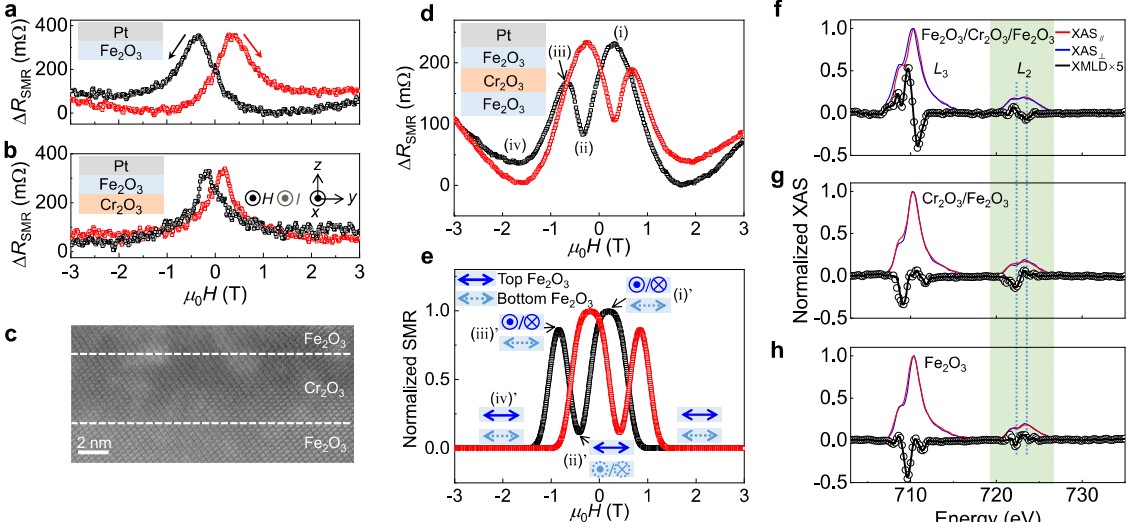

**Fig. 2 SMR and XMLD results of antiferromagnetic junctions. a**, **b** Magnetic field dependent SMR curves in control samples Fe$_2$O$_3$/Pt (**a**) and Cr$_2$O$_3$/Fe$_2$O$_3$/Pt (**b**) at 300 K. $\Delta R_{SMR}$ denotes the difference between resistance and the minimum one. Inserts are experimental set-up. **c** HAADF-STEM image of the Fe$_2$O$_3$/Cr$_2$O$_3$/Fe$_2$O$_3$ junction. **d** SMR signals of Fe$_2$O$_3$/Cr$_2$O$_3$/Fe$_2$O$_3$/Pt samples at 300 K. **e** Simulated SMR curve of Fe$_2$O$_3$/Cr$_2$O$_3$/Fe$_2$O$_3$/Pt samples at 300 K. Inserts are diagram of magnetic configurations at typical magnetic fields. **f**–**h** Normalized XAS and XMLD spectra of Fe$_2$O$_3$/Cr$_2$O$_3$/Fe$_2$O$_3$ (**f**), Cr$_2$O$_3$/Fe$_2$O$_3$ (**g**) and Fe$_2$O$_3$ (**h**) samples. The XMLD spectra were taken from the differences of XAS spectra (XAS$_\perp$–XAS$_{//}$) and then multiply by a factor of 5 at the absorption edges for clarity. The highlighted region denotes Fe-$L_2$ edge and the vertical dotted lines are guidance for eyes to mark the valleys and peaks in XMLD curves.

interlayer coupling drives the Néel vector in the top Fe$_2$O$_3$ to $n \perp H$ ($I$), again giving rise to the orthogonal configuration [(ii)' in Fig. 2e]. In this case, the spin current is reflected at the interface between Pt/top Fe$_2$O$_3$, leading to a relatively low resistance. The SMR valley in Fe$_2$O$_3$/Cr$_2$O$_3$/Fe$_2$O$_3$ occurs at the magnetic field which is much larger than that of net moment reversal (Supplementary Note 5), excluding possible coupling between net moment in Cr$_2$O$_3$ at high temperature and $n$ in Fe$_2$O$_3$ as well as artifacts due to the positive SMR from weak ferromagnetism (caused by defects or uncompensated interface). Then $n$ in the bottom Fe$_2$O$_3$ rotates towards the spin-flop state ($n \perp H$) due to the increasing negative $H$, and $n$ in the top Fe$_2$O$_3$ deviates towards $n // H$ ($I$) [(iii)' in Fig. 2e], resulting in the absorption of spin current and the second resistance peak (iii). It should be clarified that the second peak can appear when the coupling energy is large enough to overcome the Zeeman energy of the top Fe$_2$O$_3$ at the valley (ii), otherwise the $n$ (top Fe$_2$O$_3$) will maintain the spin-flop state rather than deviating towards $n // H$. The magnitude of the second peak is smaller than the first one, which can be ascribed to the less component of $n$ along the $x$-axis. In contrast, the SMR in the inverted sandwich, Fe$_2$O$_3$(4)/Cr$_2$O$_3$(4.4)/Fe$_2$O$_3$(12) (Supplementary Note 8), does not present the resistance peak before $H = 0$, demonstrating that the $n$ in the 12 nm-thick Fe$_2$O$_3$ maintains spin-flop state rather than deviating towards $H$ at low magnetic field because of the large Zeeman energy, indicating the existence of the orthogonal coupling.

Apart from magneto-transport measurements, we further confirm the interlayer coupling by direct Néel vector characterizations. Fe $L$-edge XMLD spectra were used to detect the $n$ of the top Fe$_2$O$_3$ (several nanometers-thick sensitivity) in the Fe$_2$O$_3$/Cr$_2$O$_3$/Fe$_2$O$_3$ junction, where 2 nm-thick Pt was deposited on top. The XMLD spectra were recorded at zero-field after applying a high magnetic field along the $x$-axis due to the non-volatile feature of $n$ in easy-plane Fe$_2$O$_3$[19]. X-ray was incident vertically to the film and the polarized direction was parallel to the film plane. XMLD signals are obtained as XMLD = XAS$_\perp$ − XAS$_{//}$, where XAS$_{//}$ and XAS$_\perp$ denote the x-ray absorption spectroscopy (XAS) recorded with the polarization parallel with the $x$-axis (//) and the

$y$-axis ($\perp$), respectively. Corresponding data are presented in Fig. 2f, where $L_2$-edge is highlighted because it is generally used for analyzing Fe-based XMLD spectra[34,35]. Remarkably, Fe $L_2$-edge XMLD spectrum exhibits a zero–positive–negative–zero feature, which is quite a characteristic for the $n$ along the parallel direction ($n // x$-axis)[34,35], rather than the spin-flop direction ($y$-axis). The $n$ (top Fe$_2$O$_3$) aligned along $H$ confirms the interlayer coupling, which is also corroborated by a series of XMLD measurements with sample rotation (Supplementary Note 9). In control samples Fe$_2$O$_3$ and Cr$_2$O$_3$/Fe$_2$O$_3$, where identical experiments were carried out, the scenarios differ dramatically. An opposite polarity at $L_2$-edge (Fig. 2g, h, respectively), namely zero–negative–positive–zero, was observed, suggesting that the $n$ in Fe$_2$O$_3$ is mainly aligned along the spin-flop direction ($n // y$-axis) without interlayer coupling.

**Temperature dependence of interlayer coupling**. We now turn towards the temperature dependence of SMR measurements in Fe$_2$O$_3$/Cr$_2$O$_3$/Fe$_2$O$_3$/Pt samples. Fig. 3a shows the SMR results at various temperatures (Supplementary Note 10). At a relatively high temperature ($T = 270$ K), there exists two resistance peaks as we have discussed above for $T = 300$ K, but the intensity of the second peak is lower than that at $T = 300$ K. Such a tendency continues with further decreasing temperature to 250 K, producing a tiny peak (or just a protruding), accompanied by the absence of the second peak at 200 K. It is also visible that the location of the first resistance peak shifts towards zero-field with decreasing $T$ but still maintains at positive $H$, reflecting that $n$ is parallel to $H$ in the top Fe$_2$O$_3$ before zero-field. This behavior discloses that although the interlayer coupling persists at low temperatures, the coupling energy decreases, resulting in the dominant spin-flop state and the disappearance of the second resistance peak. This phenomenon is similar to the temperature dependence of the spin fluctuation around the equilibrium position in Cr$_2$O$_3$ spacer[36]. We also demonstrate that the interlayer coupling does not depend on the direction of magnetic field and the magnitude of applied reading current (Supplementary Note 11 and 12).

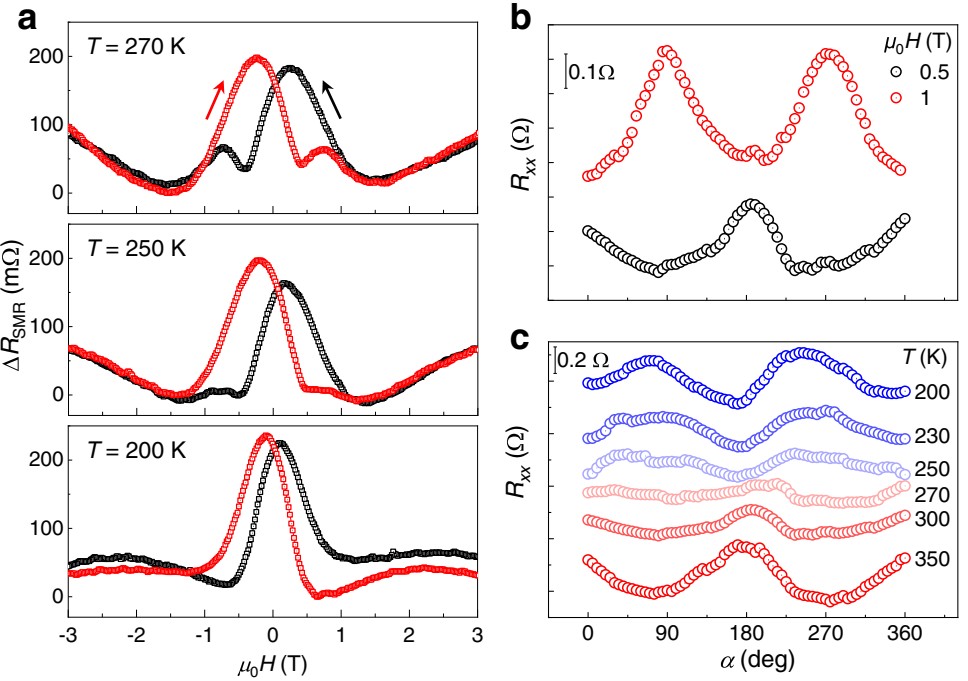

**Fig. 3 Temperature-dependent SMR signals. a** SMR signals as a function of magnetic fields for the $Fe_2O_3/Cr_2O_3/Fe_2O_3/Pt$ sample at various temperatures. **b** Angle-dependence of SMR curves for the $Fe_2O_3/Cr_2O_3/Fe_2O_3/Pt$ sample at $\mu_0H = 0.5$ and 1 T. **c** Corresponding SMR curves at various temperatures at $\mu_0H = 0.5$ T. The curves are shifted vertically for clarity.

In addition to the $H$-dependent SMR results, we also explored the interlayer coupling between antiferromagnets by in-plane angle($\alpha$)-dependent SMR measurement. Corresponding data of the $Fe_2O_3/Cr_2O_3/Fe_2O_3/Pt$ sample at $T = 300$ K measured at two typical fields of 0.5 T and 1 T are shown in Fig. 3b, where $\alpha = 0°$ means $H$ // $I$. For $\mu_0H = 1$ T, the SMR signals exhibit a negative polarity with the valley at $\alpha = 0°$, which is a typical feature for the antiferromagnetic SMR at spin-flop state[21,22]. The situation differs dramatically for $\mu_0H = 0.5$ T. The SMR curve exhibits a positive polarity, indicating that the Néel vector of the top $Fe_2O_3$ maintains $n$ // $H$ due to the dominant interlayer coupling. This finding coincides with the results of the field-dependent SMR. In contrast, the polarity of the SMR curve keeps negative in the control sample $Cr_2O_3/Fe_2O_3/Pt$ (Supplementary Note 6), reflecting the antiferromagnetic feature of $Fe_2O_3$ and the absence of the interlayer coupling. Identical angle-dependent measurements were carried out in the $Fe_2O_3/Cr_2O_3/Fe_2O_3/Pt$ sample with $\mu_0H = 0.5$ T at various temperatures. The polarity of SMR is positive at high temperatures ($T = 350$ and 300 K). When decreasing temperature to 250 K, the SMR signals become quite weak or even noisy, because of a competition between the interlayer coupling ($n$ // $H$) and the $H$-induced spin-flop ($n \perp H$). This is accompanied by the typically negative SMR induced by the spin-flop ($n \perp H$) with further decreasing temperature to 230 K and 200 K. The polarity of the control sample $Cr_2O_3/Fe_2O_3/Pt$ is always negative at different temperatures (Supplementary Note 6), reflecting the absence of the interlayer coupling and further eliminating the existence of coupling between net moment in $Cr_2O_3$ and Néel vectors in top $Fe_2O_3$.

**Analysis on the magnetic ordering.** In the following we discuss the origin of the interlayer coupling. We first consider the role of magnetic ordering in the $Cr_2O_3$ spacer. The energies related to the $Cr_2O_3$ magnetic ordering are the interfacial coupling $F(\mathbf{N}_{t,Fe}, \mathbf{M}_{t,Fe}, \mathbf{N}_{t,Cr}, \mathbf{M}_{t,Cr})$, $F(\mathbf{N}_{b,Fe}, \mathbf{M}_{b,Fe}, \mathbf{N}_{b,Cr}, \mathbf{M}_{b,Cr})$, and the magnetic energy $U$, where $\mathbf{N}$ is the Néel vector ($n$), $\mathbf{M}$ is the net

magnetization, t, b label the top and bottom surfaces, respectively. For thinner sample, the exchange energy makes it harder to let $(\mathbf{N}_{t,Cr}, \mathbf{M}_{t,Cr}) \neq (\mathbf{N}_{b,Cr}, \mathbf{M}_{b,Cr})$. Thus, the observed interlayer coupling which increases with decreasing $Cr_2O_3$ thickness cannot be explained by the non-uniform distribution of the magnetic order in $Cr_2O_3$ in the thickness direction. Also, it is known that $Cr_2O_3$ is lack of inter-unit cell DMI[32], which favors out-of-plane spiral spin structure and may cause $(\mathbf{N}_{t,Cr}, \mathbf{M}_{t,Cr}) \neq (\mathbf{N}_{b,Cr}, \mathbf{M}_{b,Cr})$. If $(\mathbf{N}_{t,Cr}, \mathbf{M}_{t,Cr}) = (\mathbf{N}_{b,Cr}, \mathbf{M}_{b,Cr})$, assuming $F(\mathbf{N}_{t,Fe}, \mathbf{M}_{t,Fe}, \mathbf{N}_{t,Cr}, \mathbf{M}_{t,Cr}) \leq F(\mathbf{N}_{b,Fe}, \mathbf{M}_{b,Fe}, \mathbf{N}_{b,Cr}, \mathbf{M}_{b,Cr})$, one can lower the total energy by rotating $(\mathbf{N}_{b,Fe}, \mathbf{M}_{b,Fe})$ towards $(\mathbf{N}_{t,Fe}, \mathbf{M}_{t,Fe})$. The process above is solid even when the interfacial coupling at different interfaces has a different magnitude, as long as the interfacial coupling has the same form. Thus, by considering the $Cr_2O_3$ magnetic ordering which is uniform in the film plane, the lowest energy state always has $(\mathbf{N}_{b,Fe}, \mathbf{M}_{b,Fe}) = (\mathbf{N}_{t,Fe}, \mathbf{M}_{t,Fe})$, i.e., no orthogonal interlayer coupling can be generated.

**Non-uniform domain wall state mediated interlayer coupling.** Having excluded the magnetic ordering which is uniform in the film plane, we consider magnetic ordering, which is non-uniform in the film plane, as the origin of the interlayer coupling. It is known that orthogonal interlayer coupling could exist in FM/NM/FM trilayers due to the interfacial roughness and oscillating collinear exchange coupling[13]. The collinear interlayer coupling mediated by antiferromagnets also oscillates as a function of the antiferromagnetic layer thickness due to the antiparallel alignment of the magnetic moments of the adjacent monolayers in the antiferromagnet[7,8]. Hence, the preferred Néel vector orientation of the top and bottom $Fe_2O_3$ could be either parallel or antiparallel because of the thickness variation of $Cr_2O_3$ layer (Supplementary Note 13). When a parallel-preferred and an antiparallel-preferred area are close enough to each other, $Fe_2O_3$ cannot form a 180° domain wall to relax the $Cr_2O_3$ magnetic order in both areas. Assuming that the Néel vector in each $Fe_2O_3$ layer is uniform, the parallel state would induce a 180° domain

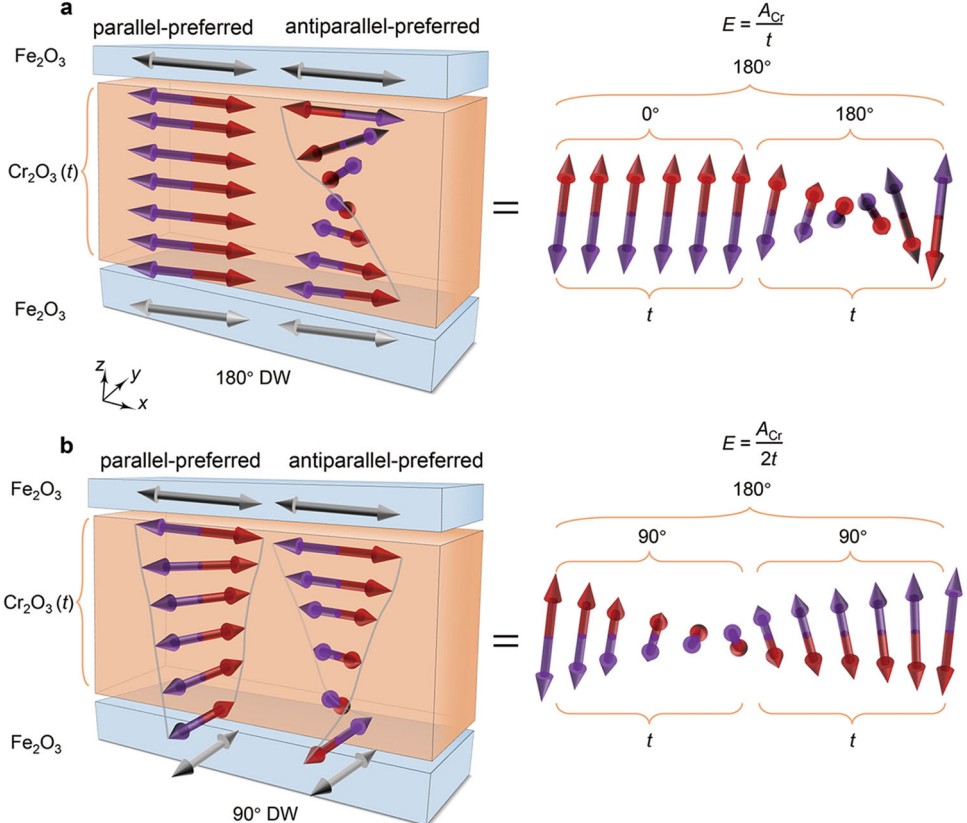

**Fig. 4 Schematic of the origin of the orthogonal interlayer coupling.** The magnetic order of $Cr_2O_3$ in parallel-preferred and antiparallel-preferred areas in the collinear state (**a**) and orthogonal state (**b**). The right insets are equivalent magnetic structure in $Cr_2O_3$. In collinear state, the 180° domain wall is induced over the $Cr_2O_3$ thickness $t$. But in the orthogonal state, the two 90° domain walls equal to a 180° domain wall over $2t$, leading to a lower energy as compared with the collinear state and the stabilization of orthogonal state. Note that the real energy of the two states are more complicated due to the relaxation of the $Fe_2O_3$ layers. $A_{Cr}$ is the exchange stiffness of $Cr_2O_3$. The gray lines are guidelines for magnetic moment rotation.

wall over the $Cr_2O_3$ thickness $t$ in the antiparallel-preferred area (Fig. 4a). The orthogonal state, however, would induce two 90° domain walls in both areas, which is equal to a 180° domain wall over $2t$ in energy (right inset of Fig. 4b). The 180° domain wall over $2t$ has lower energy than the 180° domain wall over $t$, hence the orthogonal state has lower energy, resulting in the orthogonal interlayer coupling. Considering the further relaxation of the $Fe_2O_3$ Néel vector and the distance $L$ between the parallel-preferred and the antiparallel-preferred areas, the order of the coupling energy can be estimated as $E_c \sim E_{Cr}^2/E_{Fe}$[13], where the domain wall energy in $Cr_2O_3$ is given by

$$E_{Cr} \sim \frac{A_{Cr}}{t}L^2, \tag{1}$$

and the domain wall energy in $Fe_2O_3$ is given by

$$E_{Fe} \sim \frac{A_{Fe}}{L}Lt_t. \tag{2}$$

Here, $A_{Cr}$ and $A_{Fe}$ are the exchange stiffness of $Cr_2O_3$ and $Fe_2O_3$, respectively, and $t_t$ is the thickness of the top $Fe_2O_3$. The resulting coupling energy per area reads

$$E_C \sim q\frac{A_{Cr}^2L^2}{A_{Fe}t^2t_t}, \tag{3}$$

where $q$ is the volume percentage of the in-plane Néel vector that can form this non-uniform domain wall (NUDW) state. Note that a $L^2$ factor is subtracted to get the coupling energy per area.

The maximum coupling field ($\mu_0H_{MaxCoupling}$) is inversely proportional to the square of the $Cr_2O_3$ thickness $t$,

(Supplementary Note 14) which is consistent with our model based on the NUDW state (Eq. 3).

**Interlayer coupling strength.** It is significant to characterize the interlayer coupling strength. Considering that the existence of the first peak is the compromise between the interlayer coupling and the spin-flop state, its location ($\mu_0H_{Coupling}$) as a function of temperature for different $Cr_2O_3$ thicknesses ($t$) is summarized in Fig. 5a (Supplementary Note 15) to reflect the coupling strength. The first peak persists at a positive field for all measured SMR curves, suggesting the orthogonal antiferromagnetic interlayer coupling when $t$ ranges 3–4.4 nm. The maximum coupling strength increases with decreasing $Cr_2O_3$ thickness. For thin $Cr_2O_3$ ($t = 3.0$ and $3.5$ nm), $\mu_0H_{Coupling}$ emerges from about 10 K, increases with increasing temperature and gets saturated at around 150 K. A plateau of the coupling strength exists from 150 K to 300 K. Then the coupling strength drops just above room temperature, which coincides with the spin fluctuation in $Cr_2O_3$ (bulk Néel temperature ~307 K)[36]. While for thick $Cr_2O_3$ ($t = 4.1$ and 4.4 nm), $\mu_0H_{Coupling}$ has an onset temperature of about 100 K. The $\mu_0H_{Coupling}$ increases with increasing temperature and reaches the maximum just above room temperature, then drops, without showing a plateau. The maximum coupling fields are summarized as a function of $Cr_2O_3$ thicknesses in Fig. S14 (Supplementary Note 14), which shows a consistent tendency with our NUDW model. For thicker $Cr_2O_3$ ($t = 6$ and 12 nm), $Fe_2O_3/Cr_2O_3/Fe_2O_3/$Pt samples show almost the same SMR signals at 300 K as samples with only one $Fe_2O_3$ layer (Supplementary Note 16), indicating the absence of the interlayer coupling when the $Cr_2O_3$ layer is too

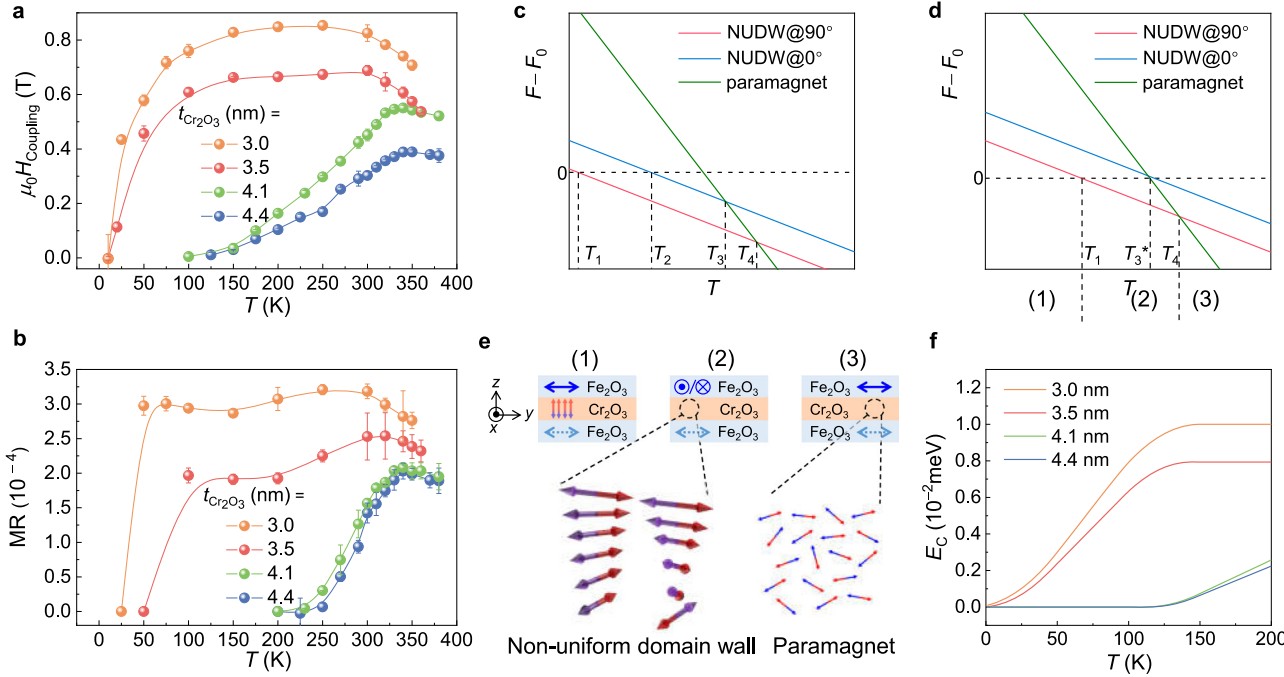

**Fig. 5 Temperature and spacer thickness dependent interlayer coupling. a** Summary of the location ($\mu_0 H_{Coupling}$) of the first peak for $Fe_2O_3$/$Cr_2O_3$/$Fe_2O_3$/Pt samples with various $Cr_2O_3$ layer thicknesses ($t = 3.0$, 3.5, 4.1, and 4.4 nm). **b** Corresponding summary of the temperature-dependent magnetoresistance (MR). The error bars are estimated from the SMR data with sweeping $H$ four times. **c, d** Schematic free energy diagrams for the thin $Cr_2O_3$ (**c**) and the thick $Cr_2O_3$ (**d**) cases. **e** Magnetic order in $Fe_2O_3$/$Cr_2O_3$/$Fe_2O_3$ at three temperature ranges (1)–(3) as marked in **d**. **f** Temperature dependence of calculated coupling energy.

thick. In addition, the interlayer coupling is observed in the $Fe_2O_3$/$NiO$/$Fe_2O_3$ junction with antiferromagnetic NiO spacer (Supplementary Note 17), indicating that the coupling effect is not restricted to a certain spaced material.

The distinct behavior for the samples with thin and thick $Cr_2O_3$ shows a significant role of spacer thickness. Since the out-of-plane anisotropy decrease rapidly with smaller sample size[37], the most possible origin of the two distinct types of temperature dependence is that at low temperature, the Néel vectors in the thin $Cr_2O_3$ already have large in-plane component[37–39], but the thick $Cr_2O_3$ have stable out-of-plane Néel vector. The coupling arises from the fluctuating magnetic moments, hence thin $Cr_2O_3$ mediates coupling at low temperature, and thick $Cr_2O_3$ can only mediate coupling above 100 K.

As the coupling field shifts towards $H = 0$ with decreasing temperature, the magnitude of the second resistance peak, which is also related to the deviation of Néel vector in top $Fe_2O_3$, also changes (Fig. 3a). We then summarize the temperature-dependent magnetoresistance (MR) for the $Fe_2O_3$/$Cr_2O_3$/$Fe_2O_3$/Pt samples with various $t$ in Fig. 5b. MR is related to the magnitude of the second resistance peak, and is defined as MR = [$R$(second peak)–$R$(lowest)]/$R$(lowest), where the lowest is the minimum of the SMR curves. It can be seen that the MR curve exhibits a similar temperature dependence as the coupling field, strongly suggesting that the MR is also relevant to the interlayer coupling. As we discussed above (Fig. 2d), the second resistance peak is the result of competition between interlayer coupling and spin-flop state in the top $Fe_2O_3$. When the temperature is low, the coupling energy is relatively small as compared with the Zeeman energy. Therefore, the Néel vector in the top $Fe_2O_3$ maintains spin-flop state ($n \perp H$), causing the vanishment of the second peak. The vanishing temperature (MR = 0) is obviously higher than its counterpart where coupling effect disappears ($\mu_0 H_{Coupling} = 0$), such as $T = 225$ K and $T = 125$ K for $t = 4.4$ nm, respectively. As the temperature increases further, the coupling energy is enhanced and

exceeds the Zeeman energy, resulting in the deviation of more $n$ towards $H$ and the resultant rapid rise of the second resistance peak (MR). As the spin correlation is partially destroyed at high temperatures ($T > 340$ K for $t = 4.4$ nm), the coupling is reduced, accompanied by the decreasing of MR. An analogical situation occurs for the other samples ($t = 3.0$, 3.5, and 4.1 nm), while the samples with thin ($t = 3.0$ and 3.5 nm) and thick ($t = 4.1$ and 4.4 nm) $Cr_2O_3$ are divided into two groups according to the temperature dependence of the coupling strength.

The unique temperature dependence of the orthogonal interlayer coupling strength in the antiferromagnetic trilayers is dramatically different from its counterpart in ferromagnet/normal metal/ferromagnet trilayers, which is insensitive to temperature[10]. We attribute this temperature dependence to the evolution of the magnetic order of $Cr_2O_3$ caused by the temperature-dependent anisotropy. The schematic free energy diagrams at zero field for the thin and thick $Cr_2O_3$ cases are shown in Fig. 5c, d, respectively. The out-of-plane ground state has a free energy of $F_0$, and all other states are represented by their free energy difference with the ground state $F - F_0$. In thin $Cr_2O_3$ cases, no coupling exists in the ground state. At $T_1$, in-plane NUDW state emerges under the perpendicular condition, while the $Cr_2O_3$ under the parallel condition remains in the ground state. Therefore, the coupling emerges with a coupling strength represented by $F_0 - F$(NUDW@90°) and increasing from $T_1$ to $T_2$. At $T_2$, the $Cr_2O_3$ under the parallel condition switches from the ground state to the NUDW state, and the coupling strength saturates to $F$(NUDW@0°)–$F$(NUDW@90°). Then the $Cr_2O_3$ under the parallel condition switches from the NUDW state to paramagnetic state at $T_3$, and the coupling strength $F$(paramagnet) − $F$(NUDW@90°) starts to decrease. The coupling strength finally vanishes at $T_4$, where the $Cr_2O_3$ under the perpendicular condition switches to the disorder paramagnetic phase. Similar process occurs in the thick $Cr_2O_3$ cases, with a larger $T_1$ because the perpendicular anisotropy stabilizes the

Néel order in the ground state. The coupling strength increases up to $T_3^*$, where the $Cr_2O_3$ under the parallel condition switches directly from the ground state to the disorder state, without entering the NUDW state. Then the coupling strength decreases and finally vanishes at $T_4$.

The phase transition from the out-of-plane ground state to the NUDW state can be phenomenally described by the following free energy

$$F = (\Delta - \lambda T)q + bn^4 \qquad (4)$$

where $T$ is the temperature, $b$ is a parameter to stabilize the out-of-plane Néel Order, and $\lambda$ is a parameter related to the entropy difference between the out-of-plane ground state and the in-plane NUDW. $\Delta$ reflects the energy difference between the ground state and the NUDW, with values $\Delta_\perp$ and $\Delta_{//}$ for the perpendicular and the parallel conditions, respectively. $\Delta_\perp$ is smaller than $\Delta_{//}$ and $\Delta_{//} - \Delta_\perp$ increases with decreasing $Cr_2O_3$ thickness. The out-of-plane Néel Order $n$ and the volume of the NUDW state $q$ satisfies

$$q + n^2 = 1 \qquad (5)$$

The first term in (4) describes the blue and red lines in Fig. 5c, d, and a combination of (4) and (5) gives the $n$-related part of the free energy $\sim (\Delta/\lambda - T) \; n^2 + b \; n^4$, yielding a characteristic temperature for the spin-reorientation transition $\Delta/\lambda$ and $n \sim \sqrt{\Delta/\lambda - T}$ near this temperature[40]. The coupling energy $E_c$, the free energy difference under the two conditions (Supplementary Note 18), is displayed in Fig. 5f, which qualitatively agrees with the experimental curves ($\leq$200 K).

## Discussion

Both the experimental and theoretical results disclose that a small energy difference between the parallel and perpendicular states can be embodied as sizable interlayer coupling fields for the antiferromagnetic interlayer coupling owing to the vanishingly small net moment, exhibiting unique advantage as compared with its ferromagnetic counterpart. A combination of the temperature and spacer thickness dependent SMR measurements, XMLD characterizations and the theory model demonstrates the orthogonal interlayer coupling in antiferromagnetic junctions.

In summary, the present discovery of strong orthogonal interlayer coupling in the all-antiferromagnetic junction exceeds the category of traditional collinear interlayer coupling, and is proposed to be mediated by the non-uniform domain wall state in spacer. In addition, other magnetic states such as Néel vector fluctuation induced magnon may also exist in our system and mediate the orthogonal interlayer coupling[41]. Such an orthogonal interlayer coupling in AFMs would open a new avenue for noncollinear coupling in condensed matter and hopefully serve as a promising basic building block for functional antiferromagnetic devices aiming at data processing and storage with ultrahigh-density integration and ultrafast speed[14,15].

## Methods

**Sample preparations**. The all-antiferromagnetic junctions $Fe_2O_3$(12 nm)/$Cr_2O_3$($t$ nm)/$Fe_2O_3$ (4 nm) ($t$ = 3.0, 3.5, 4.1, and 4.4 nm) and control samples $Fe_2O_3$(12 nm), $Cr_2O_3$(4.4 nm)/$Fe_2O_3$(4 nm) were deposited on $Al_2O_3$ (0001) substrates in pulse laser deposition (PLD) system at 873 K, with a base vacuum of $1 \times 10^{-8}$ torr. Then a 4 nm platinum layer was covered on the junctions by direct current sputtering at room temperature for spin Hall magnetoresistance (SMR) measurements to detect the orientation of Néel vectors. The highly insulating characteristic of samples is confirmed by resistivity measurements (Supplementary Note 1).

**SMR measurements**. The junctions were fabricated into Hall bars by standard photolithography combined with argon ion etching. A Keithley 2400 instrument provided a current $I$ along the $x$-axis in the platinum for SMR measurements. A Keithley 2182 instrument was used to record the voltage along the platinum stripe during the magnetic field dependence of SMR measurement. The magnetic field ($H$) was applied along the same direction (along the $x$-axis). The in-plane angle

dependence of SMR is performed in a physical property measurement system (PPMS, Quantum Design).

**XMLD measurements**. X-ray magnetic linear dichroism (XMLD) spectra were carried out at Beamline 08U1A of the Shanghai Synchrotron Radiation Facility (SSRF). The x-ray absorption spectroscopy (XAS) data were obtained in total electron yield mode, which reflects the electronic structure of top $Fe_2O_3$ within several nanometers, therefore the bottom $Fe_2O_3$ cannot contribute to XAS. A 2 nm-thick Pt capping layer was deposited on samples for electron conduction. The Fe $L$-edge XMLD spectra were obtained by the difference between linearly horizontal (//) and vertical ($\perp$) polarized XAS.

## Data availability

The data that support the findings of this study are available from the corresponding authors upon reasonable request.

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

## Acknowledgements
We thank D.Z. Hou, R. Cheng, J. Xiao, X.G. Wan, K. Shen, Y.Z. Wu, and P. Yan for fruitful discussion. This work was supported by the National Key Research and Development Program of China (MOST) (Grant No. 2021YFB3601301), the National Natural Science Foundation of China (Grant No. 51871130), and the Natural Science Foundation of Beijing, China (Grant No. JQ20010). We thank Beamline 08U1A of SSRF for XMLD measurements.

## Author contributions
C.S. led the project. Y.Z. and C.S. proposed the study. Y.Z. prepared the samples and carried out the measurements with the help from H.B. and M.Z.L.L. and T.G. conducted theoretical analysis. Y.Z., L.L. and C.S. wrote the manuscript. All authors discussed the results and commented on the manuscript.

## Competing interests
The authors declare no competing interests.
