## [Peer Review File · Nature Communications]

Reviewers' Comments:

Reviewer #1:

Remarks to the Author:

The manuscript is devoted to the interlayer coupling between two antiferromagnetic (AFM) Fe₂O₃ layers across a thin layer of insulating antiferromagnet Cr₂O₃. The authors observe a spin Hall magnetoresistance (SMR) effect exhibiting features which indicate the orthogonal alignment of the Néel vectors of the two Fe₂O₃ layers in the ground state. This interpretation is supported by their x-ray magnetic linear dichroism (XMLD) data. The authors argue that they observe an orthogonal interlayer coupling driven by a quasi-long range order in the AFM spacer layer.

Historically, the interlayer coupling has been a hot topic of research since its discovery in 1986 by P. Grünberg et al., *Phys. Rev. Lett.* 57, 2442 (1986), due to its relevance to the giant magnetoresistance (GMR) effect discovered later. This bilinear ($\sim m_1 \cdot m_2$) coupling across a metallic spacer layer, oscillating as a function of its thickness, is now well understood within a model of the standing wave in the metallic spacer layer (e.g., M. D. Stiles, *Interlayer exchange coupling*, in *Ultrathin Magnetic Structures III* p. 99, (2005)). In addition to the bilinear coupling, a biquadratic ($\sim (m_1 \cdot m_2)^2$) coupling has been discovered (see, e.g., S. O. Demokritov, *J. Phys. D: Appl. Phys.* 31, 925 (1998), for review). While the bilinear coupling aligns the magnetic moments collinear (parallel or antiparallel), the biquadratic coupling tends to align the moments orthogonal. The widely accepted explanation of the biquadratic coupling is Slonczewski's model (J. C. Slonczewski, *Phys. Rev. Lett.* 67, 3172 (1991)), where the interface roughness in conjunction with the oscillatory bilinear coupling drives the effect.

In addition to the coupling across metallic spacers, the interlayer coupling across non-magnetic (e.g., T. Katayama et al., *Appl. Phys. Lett.* 89, 112503 (2006)) and antiferromagnetic (e.g., Z. Y. Liu and S. Adenwalla, *Phys. Rev. Lett.* 91, 037207 (2003)) insulators has been observed. The latter oscillates as a function of the antiferromagnetic layer thickness due to the antiparallel alignment of the magnetic moments of the adjacent monolayers in the antiferromagnet (M. Y. Zhuravlev et al., *Phys. Rev. Lett.* 92, 219703 (2004)).

The submitted manuscript addresses the interlayer coupling between antiferromagnetic layers, which distinguishes this work from the previous studies where the interlayer coupling was observed between ferromagnetic layers. In my view, the authors provided compelling arguments based on their SMR and XMLD data that the Néel vectors of the two Fe₂O₃ layers are orthogonal in the ground state. These results are interesting and should be published in some form.

Having said this, I am not convinced at all that their model of a quasi-long range order in the antiferromagnetic spacer layer correctly explains their results. While the Cr₂O₃ (0001) moments are orthogonal to the interfaces, they are capable to transmit the exchange coupling between the two Fe₂O₃ layers. While this coupling is expected to be bilinear (despite the likely small canting of the magnetic moments), the presence of roughness can produce a biquadratic coupling according to Slonczewski's mechanism. This explanation seems to me more plausible than that provided by the authors.

There are few other comments which need to be taken into account by the authors.

1. The authors need to place their work properly in the context of the previous knowledge in the field. The references given above my serve as a guide to the authors.

2 The manuscript needs to be significantly improved in terms of the English and the style. There are many unclear or incorrect statements. Below I list just a couple of them in the abstract.

- The first sentence in the abstract is erroneous and misleading.

- The third sentence in the abstract is unclear.

- The fifth sentence in the abstract is incorrect (see comment 3).

3. In the last sentence on page 2, the authors say that "collinear parallel/antiparallel arrangements

of Néel vectors in antiferromagnets are identical." They are not identical by symmetry. It is not clear what the authors want to say. Also, in the same sentence they say "the small noncollinear interaction would become dominant in an AFM/spacer/AFM junction." The bilinear interlayer coupling occurs across an AFM layer. This interaction can cause small canting of the magnetic moments, but it is still bilinear.

4. In their model, the authors introduce the Dzyaloshinskii-Moriya interaction (DMI) in Fe₂O₃. It is not clear what its role is. On page 4, they say that "the hysteresis is due to the existence of Dzyaloshinskii-Moriya interaction (DMI)..." However, normally the hysteresis can be well explained by the magnetic anisotropy...

5. On page 5, the authors say; "The resistance peak appears before $H = 0$, which violates the principle of thermodynamics, indicating the existences of coupling effect." It is not clear why the principle of thermodynamics is violated in their system. Probably, the authors meant something else.

Reviewer #3:

Remarks to the Author:

The manuscript by Zhou et al. reports an orthogonal coupling of two Fe₂O₃ layers separated by a Cr₂O₃ layer. Firstly, the results of magneto-transport measurements on Fe₂O₃/Cr₂O₃/Fe₂O₃/Pt structure as well as two reference samples with only Fe₂O₃/Pt layers and Cr₂O₃/Fe₂O₃/Pt layers are presented. The results are well explained by assuming an emergence of an orthogonal coupling for the two Fe₂O₃ layers. XMLD results support this picture. Then, a temperature dependence of the magneto-transport properties is presented, from which the origin of the orthogonal coupling is discussed. Finally, a theoretical model is put forward, with which the experiment is semi-quantitatively explained.

The interlayer magnetic coupling is an important phenomenon for functional magnetic and spintronic devices as well as a fundamental interest in condensed-matter physics, and collinear antiferromagnetic coupling is widely used in hard-disc drive and nonvolatile memory. Meanwhile, as far as I know, the orthogonal interlayer coupling has not been well studied so far. In this regard, I think this work would potentially give a high impact to the community of magnetics and spintronics. However, I think the authors should consider the following three points and revise the manuscript.

(1) The authors assume that the signal of the magneto-transport measurement arises from the spin-Hall magnetoresistance (SMR), but its evidence is not presented in this manuscript. For example, what about the contribution of anisotropic magnetoresistance and other magnetoresistive effects. I think the authors should perform the same measurement for samples without the Pt cap layer or with a cap layer comprised by a low spin Hall material.

(2) The authors attribute the orthogonal coupling to a quasi-long range order (QLRO), but I could not find any solid evidence of the QLRO in Cr₂O₃ layer. I think the authors should somehow investigate the spin structure of the Cr₂O₃ layer as a function of the temperature and/or the thickness of Cr₂O₃ layer; otherwise, the proposed scenario is just a speculation. In addition, I cannot understand well why the orthogonal coupling between the Fe₂O₃ is stabilized when the QLRO emerges in Cr₂O₃. More detailed explanation is desirable.

(3) English should be improved. There are many sentences containing grammatical error.

Response Letter of NCOMMS-21-38921A-Z

We very much appreciate the positive evaluations of our manuscript (NCOMMS-21-38921A-Z) by Reviewer #1 (“In my view, the authors provided compelling arguments based on their SMR and XMLD data” and “These results are interesting and should be published in some form”) and Reviewer #3 (“I think this work would potentially give a high impact to the community of magnetism and spintronics”). We address the issues raised by them point by point below. And their comments are helpful for our improvements further. Leilei Qiao and Yunfeng You are added as coauthor for the help in the physical model and English writing improvement. Amendments of our revised manuscript are summarized below in bold face style.

The main revisions include:

1. We revised our theory model as suggested by Reviewer #1.
2. We revised the statements in the abstract, introduction as well as the main text as pointed out by Reviewer #1.
3. We performed magnetoresistance measurement for samples with a cap layer comprised by a low spin Hall material titanium (Ti) as suggested by Reviewer #3.
4. We added more detailed explanation about our theory as pointed out by Reviewer #3.
5. We carefully revised the English writing throughout the manuscript.

Response to Reviewer #1

The manuscript is devoted to the interlayer coupling between two antiferromagnetic (AFM) Fe_2O_3 layers across a thin layer of insulating antiferromagnet Cr_2O_3 . The authors observe a spin Hall magnetoresistance (SMR) effect exhibiting features which indicate the orthogonal alignment of the Néel vectors of the two Fe_2O_3 layers in the ground state. This interpretation is supported by their x-ray magnetic linear dichroism (XMLD) data. The authors argue that they observe an orthogonal interlayer coupling driven by a quasi-long range order in the AFM spacer layer.

Historically, the interlayer coupling has been a hot topic of research since its discovery in 1986 by P. Grünberg et al., Phys. Rev. Lett. 57, 2442 (1986), due to its relevance to the giant magnetoresistance (GMR) effect discovered later. This bilinear ($\sim m_1 \cdot m_2$) coupling across a metallic spacer layer, oscillating as a function of its

thickness, is now well understood within a model of the standing wave in the metallic spacer layer (e.g., M. D. Stiles, Interlayer exchange coupling, in *Ultrathin Magnetic Structures III* p. 99, (2005)). In addition to the bilinear coupling, a biquadratic ($\sim(m_1 \cdot m_2)^2$) coupling has been discovered (see, e.g., S. O. Demokritov, *J. Phys. D: Appl. Phys.* 31, 925 (1998), for review). While the bilinear coupling aligns the magnetic moments collinear (parallel or antiparallel), the biquadratic coupling tends to align the moments orthogonal. The widely accepted explanation of the biquadratic coupling is Slonczewski's model (J. C. Slonczewski, *Phys. Rev. Lett.* 67, 3172 (1991)), where the interface roughness in conjunction with the oscillatory bilinear coupling drives the effect.

In addition to the coupling across metallic spacers, the interlayer coupling across non-magnetic (e.g., T. Katayama et al., *Appl. Phys. Lett.* 89, 112503 (2006)) and antiferromagnetic (e.g., Z. Y. Liu and S. Adenwalla, *Phys. Rev. Lett.* 91, 037207 (2003)) insulators has been observed. The latter oscillates as a function of the antiferromagnetic layer thickness due to the antiparallel alignment of the magnetic moments of the adjacent monolayers in the antiferromagnet (M. Y. Zhuravlev et al., *Phys. Rev. Lett.* 92, 219703 (2004)).

The submitted manuscript addresses the interlayer coupling between antiferromagnetic layers, which distinguishes this work from the previous studies where the interlayer coupling was observed between ferromagnetic layers. In my view, the authors provided compelling arguments based on their SMR and XMLD data that the Néel vectors of the two Fe₂O₃ layers are orthogonal in the ground state. These results are interesting and should be published in some form.

A: We sincerely thank the Reviewer for the positive evaluation such as “the authors provided compelling arguments”, “These results are interesting and should be published in some form” and recommendation for the publication of our paper. We have addressed the Reviewer's comments as shown below.

Q1) Having said this, I am not convinced at all that their model of a quasi-long range order in the antiferromagnetic spacer layer correctly explains their results. While the Cr₂O₃ (0001) moments are orthogonal to the interfaces, they are capable to transmit the exchange coupling between the two Fe₂O₃ layers. While this coupling is expected to be bilinear (despite the likely small canting of the magnetic moments), the presence of roughness can produce a biquadratic coupling according to Slonczewski's mechanism. This explanation seems to me more plausible than that provided by the authors.

A: Thanks for the helpful suggestion from the Reviewer. We revised our model based on Slonczewski's mechanism and we found that the interfacial roughness plays an important role in forming the orthogonal interlayer coupling.

Based on this analysis, we think that the quasi-long range order state in the Cr_2O_3 spacer may be better described as an in-plane non-uniform domain wall state. In fact, this picture is still similar to our previous model, in that there is in-plane magnetic order in the Cr_2O_3 spacer, and this magnetic order is non-uniform in the film plane. The moments form magnetic order in a short range, but the whole Cr_2O_3 spacer cannot be described by a single Néel vector. Meanwhile, the original Slonczewski model in ferromagnet/normal metal/ferromagnet is insensitive to temperature, which cannot be used to describe the unique temperature dependence of the coupling in our antiferromagnetic trilayer. The temperature evolution of the magnetic order in the Cr_2O_3 spacer is an additional degree of freedom that has to be taken into account. Hence, we think that the combination of the Slonczewski model and our previous model that focus on the temperature evolution of the magnetic order in the Cr_2O_3 spacer provides the thorough description for the coupling phenomena.

We rewrote the modelling part on Page 11 as following:

“Non-uniform domain wall state mediated interlayer coupling.

Having excluded the magnetic ordering which is uniform in the film plane, we consider magnetic ordering, which is non-uniform in the film plane, as the origin of the interlayer coupling. It is known that orthogonal interlayer coupling could exist in FM/NM/FM trilayers due to the interfacial roughness and oscillating collinear exchange coupling¹³. The collinear interlayer coupling mediated by antiferromagnets also oscillates as a function of the antiferromagnetic layer thickness due to the antiparallel alignment of the magnetic moments of the adjacent monolayers in the antiferromagnet^{7,8}. Hence, the preferred Néel vector orientation of the top and bottom Fe_2O_3 could be either parallel or antiparallel. When a parallel-preferred and an antiparallel-preferred area are close enough to each other, the Fe_2O_3 cannot form a 180° domain wall to relax the Cr_2O_3 magnetic order in both areas. Assuming that the Néel vector in each Fe_2O_3 layer is uniform, the parallel state would induce a 180° domain wall over the Cr_2O_3 thickness t in the antiparallel-preferred area (Fig. 4a). The orthogonal state, however, would induce two 90° domain walls in both areas, which is equal to a 180° domain wall over $2t$ in energy. The 180° domain wall over $2t$ has a lower energy than the 180°

domain wall over t , hence the orthogonal state has a lower energy, resulting in the orthogonal interlayer coupling. Considering the further relaxation of the Fe_2O_3 Néel vector and the distance L between the parallel-preferred and the antiparallel-preferred areas, the order of the coupling energy can be estimated as $E_c \sim E_{\text{Cr}}^2/E_{\text{Fe}}$,¹³ where the domain wall energy in Cr_2O_3 is given by

$$E_{\text{Cr}} \sim \frac{A_{\text{Cr}}}{t} L^2, \quad (1)$$

and the domain wall energy in Fe_2O_3 is given by

$$E_{\text{Fe}} \sim \frac{A_{\text{Fe}}}{L} L t_t. \quad (2)$$

Here, A_{Cr} and A_{Fe} are the exchange stiffness of Cr_2O_3 and Fe_2O_3 , respectively, and t_t is the thickness of the top Fe_2O_3 . The resulting coupling energy per area reads

$$E_c \sim q \frac{A_{\text{Cr}}^2 L^2}{A_{\text{Fe}} t^2 t_t}, \quad (3)$$

where q is the volume percentage of the in-plane Néel vector that can form this NUDW state. Note that a L^2 factor is subtracted to get the coupling energy per area.

The maximum coupling field ($\mu_0 H_{\text{MaxCoupling}}$) is inversely proportional to the square of the Cr_2O_3 thickness t , (Supplementary Note 13) which is consistent with our model based on the non-uniform domain wall (NUDW) state (Eq. 3).

Fig. 4 Schematic of the origin of the orthogonal interlayer coupling. The magnetic order of Cr_2O_3 in parallel-preferred and antiparallel-preferred areas in the collinear state (a) and orthogonal state (b)."

We added the plot of coupling field as a function of Cr_2O_3 thickness t to

Supplementary Note 13:

Note 13. Maximum coupling field as a function of Cr_2O_3 thickness

To further investigate the coupling effect and the spin structure in Cr_2O_3 , the maximum coupling field ($\mu_0 H_{\text{MaxCoupling}}$) as a function of Cr_2O_3 thickness is plotted. The maximum coupling field is inversely proportional to the square of the Cr_2O_3 thickness t , (Fig. S14) which is consistent with our model based on the non-uniform domain wall state (Eq. 3 in the main text). Such a consistency verifies the proposed spin structure in Cr_2O_3 in the non-uniform domain wall model.

Fig. S14 Maximum coupling field ($\mu_0 H_{\text{MaxCoupling}}$) as a function of the square of the Cr_2O_3 thickness t . The maximum coupling field is inversely proportional to the square of t .

We added sentences on Page 16, Line 1:

“The unique temperature dependence of the orthogonal interlayer coupling strength in the antiferromagnetic trilayers is dramatically different from its counterpart in ferromagnet/normal metal/ferromagnet trilayers, which is insensitive to temperature¹⁰. We attribute this temperature dependence to the evolution of the magnetic order of Cr_2O_3 caused by the temperature-dependent anisotropy.”

We added sentences at Supplementary Note 17:

“We next compare this energy with Eq. (3) in the main text. Since the Néel temperature of Fe_2O_3 (956 K)^{S13} is approximately three times of that of Cr_2O_3 (307 K)^{S14}, we use $A_{\text{Fe}} \approx 3 A_{\text{Cr}}$ in the estimation. Considering $L \approx t$, $t_t = 18 a$, where a is the monolayer distance, and the exchange coefficient $J = 8$ meV for the Cr_2O_3 ^{S15}, $A_{\text{Cr}} \approx Ja$, we get $E_c \sim 0.049$ meV (per unit cell), in the same order compared with the estimated value from the experiment. The relatively smaller experimental value could be due to the relaxation in the Fe_2O_3 in the out-of-plane direction, and the fact that the

interfacial Fe and Cr moments are not identically parallel or antiparallel^{S16}.”

There are few other comments which need to be taken into account by the authors.

Q2) The authors need to place their work properly in the context of the previous knowledge in the field. The references given above may serve as a guide to the authors.

A: The Reviewer's comment is reasonable. We revised the sentences at the beginning of our manuscript to cover previous knowledge in the research field as the Reviewer pointed out. **“The most well-established example is the giant magnetoresistance system, ferromagnet/transition metal/ferromagnet¹⁻⁵, where the electron standing wave state⁶ in spacer is induced by the magnetizations in two ferromagnets (FMs) and leads to the collinear interlayer coupling. Apart from metallic spacer, the interlayer coupling can exist across antiferromagnetic^{7,8} and non-magnetic⁹ insulators, providing more material alternatives for devices. Noncollinear coupling may also exist in FMs/spacer/FMs¹⁰⁻¹², which is caused by interface roughness and the oscillatory collinear coupling¹³. But such a coupling is usually overshadowed by collinear coupling¹⁰, which is much easier for detection than the noncollinear coupling in FMs.”**

Q3) The manuscript needs to be significantly improved in terms of the English and the style. There are many unclear or incorrect statements. Below I list just a couple of them in the abstract.

- The first sentence in the abstract is erroneous and misleading.
- The third sentence in the abstract is unclear.
- The fifth sentence in the abstract is incorrect (see comment 3).

A: We have carefully revised the English writing throughout the whole manuscript.

We revised the first sentence in the abstract **“The collinear interlayer coupling between magnetic moments in ferromagnet/spacer/ferromagnet sandwich is widely studied and accelerates the development of spintronics, while the noncollinear coupling is usually absent because of the low coupling energy and large magnetization in ferromagnets.”**

The coupling effect can be detected as a magnetic field where the coupling energy equals Zeeman energy ($M \cdot H$). Therefore, in antiferromagnets with a

small net moment, the low coupling energy can be reflected as a large effective magnetic field. We modified the third sentence **“However, the small net moment in AFMs can embody a low coupling energy as a sizable coupling field, enabling the detectability of noncollinear interlayer coupling.”**

As the Reviewer pointed out in comment 3, the AFM spacer can mediate interlayer coupling between two ferromagnets. However, this is not our case here. In our all-AFM junction, the uniform magnetic ordering (net moment) in Cr_2O_3 cannot generate orthogonal interlayer coupling (“Analysis on the magnetic ordering” section). In addition, the net moment in Cr_2O_3 mediated coupling is also excluded by comparison between SMR and *M-H* loop. We revised the fifth sentence **“From the energy and symmetry analysis, the direct coupling via uniform magnetic ordering in Cr_2O_3 spacer in our junction here is excluded.”**

Q4) In the last sentence on page 2, the authors say that “collinear parallel/antiparallel arrangements of Néel vectors in antiferromagnets are identical.” They are not identical by symmetry. It is not clear what the authors want to say. Also, in the same sentence they say “the small noncollinear interaction would become dominant in an AFM/spacer/AFM junction.” The bilinear interlayer coupling occurs across an AFM layer. This interaction can cause small canting of the magnetic moments, but it is still bilinear.

A: The Reviewer’s comment is reasonable. In ideal conditions, the collinear parallel/antiparallel arrangements of Néel vectors in antiferromagnets are not identical. But in multi-domain state, such arrangements of Néel vectors are identical. In addition, the collinear arrangements (parallel and antiparallel) of Néel vectors usually cannot be detected by resistance owing to the 180° period of magnetoresistance. Therefore, the collinear arrangements are usually identical and cannot be distinguished in resistance measurement.

In contrast, the noncollinear coupling of Néel vectors can be readout in magnetoresistance measurement because of the different resistance states. Thus the noncollinear interaction can be detected in an AFM/spacer/AFM junction, although the coupling energy is not very large.

Accordingly, we revised the sentence to eliminate ambiguity **“Moreover, collinear parallel/antiparallel arrangements of Néel vectors in antiferromagnets are usually identical in magnetoresistance measurements, so that the small noncollinear interaction can be clearly**

detected in an AFM/spacer/AFM junction.”

Q5) In their model, the authors introduce the Dzyaloshinskii-Moriya interaction (DMI) in Fe₂O₃. It is not clear what its role is. On page 4, they say that “the hysteresis is due to the existence of Dzyaloshinskii-Moriya interaction (DMI)...” However, normally the hysteresis can be well explained by the magnetic anisotropy...

A: As the Reviewer pointed out, the hysteresis in our sample can be explained by Zeeman energy and anisotropy energy. The strong Dzyaloshinskii-Moriya interaction (DMI) causes the canting of Néel vector and induces a net moment in Fe₂O₃. The hysteretic behavior suggests that the DMI induced net moment is 180° switched when the Zeeman energy overcomes the anisotropy energy. The Néel vector, which is locked to the DMI induced net moment, is also 180° switched, leading to the observed SMR hysteresis loop. Without this DMI induced net moment, when sweeping the magnetic field, the Néel vector would not be 180° switched, but rather be canted to opposite direction under opposite magnetic field, so cannot generate an SMR hysteresis loop. Therefore, the DMI is introduced into our model.

Accordingly, we revised the sentences **“The SMR signals of the control samples are simulated and shown in Supplementary Note 2, where the hysteresis is caused by the competition between Zeeman energy and anisotropy energy. The existence of Dzyaloshinskii-Moriya interaction (DMI) in Fe₂O₃³² induces canting moment and the resultant switching hysteresis behavior.”**

Q6) On page 5, the authors say; “The resistance peak appears before $H = 0$, which violates the principle of thermodynamics, indicating the existences of coupling effect.” It is not clear why the principle of thermodynamics is violated in their system. Probably, the authors meant something else.

A: Generally speaking, the resistance peak, which means the switching of magnetic ordering, should appear after $H = 0$ (hysteresis behavior). This is decided by the energy competition between Zeeman energy and anisotropy energy. Therefore, the magnetic ordering switching phenomenon occurs before $H = 0$ suggests that there should exist another effect contributing to the system energy. Accordingly, we revised the sentence **“One resistance peak appears before $H = 0$, indicating the existence of coupling effect.”** to eliminate misunderstanding.

Response to Reviewer #3

The manuscript by Zhou et al. reports an orthogonal coupling of two Fe₂O₃ layers separated by a Cr₂O₃ layer. Firstly, the results of magneto-transport measurements on Fe₂O₃/Cr₂O₃/Fe₂O₃/Pt structure as well as two reference samples with only Fe₂O₃/Pt layers and Cr₂O₃/Fe₂O₃/Pt layers are presented. The results are well explained by assuming an emergence of an orthogonal coupling for the two Fe₂O₃ layers. XMLD results support this picture. Then, a temperature dependence of the magneto-transport properties is presented, from which the origin of the orthogonal coupling is discussed. Finally, a theoretical model is put forward, with which the experiment is semi-quantitatively explained.

The interlayer magnetic coupling is an important phenomenon for functional magnetic and spintronic devices as well as a fundamental interest in condensed-matter physics, and collinear antiferromagnetic coupling is widely used in hard-disc drive and nonvolatile memory. Meanwhile, as far as I know, the orthogonal interlayer coupling has not been well studied so far. In this regard, I think this work would potentially give a high impact to the community of magnetics and spintronics. However, I think the authors should consider the following three points and revise the manuscript.

A: We very much appreciate the Reviewer's positive evaluation such as "The interlayer magnetic coupling is an important phenomenon" and "I think this work would potentially give a high impact to the community of magnetics and spintronics". We have addressed issues pointed out by the Reviewer below.

Q1) The authors assume that the signal of the magneto-transport measurement arises from the spin-Hall magnetoresistance (SMR), but its evidence is not presented in this manuscript. For example, what about the contribution of anisotropic magnetoresistance and other magnetoresistive effects. I think the authors should perform the same measurement for samples without the Pt cap layer or with a cap layer comprised by a low spin Hall material.

A: Thanks for the Reviewer's helpful suggestion. To clearly verify that the SMR contribute to the signal, we performed measurement with a cap layer comprised by a low spin Hall material titanium (Ti), as the Reviewer suggested. The corresponding data are added to Supplementary Note 4.

Note 4. Magnetoresistance measurement with low spin Hall material as the cap layer.

In order to verify that the magnetoresistance is caused by SMR, we

performed measurement with the cap layer comprised by a low spin Hall material titanium (Ti) under the same test condition with that in $\text{Fe}_2\text{O}_3/\text{Cr}_2\text{O}_3/\text{Fe}_2\text{O}_3/\text{Pt}$. The corresponding data are shown in Fig. S4. It can be seen that there exists negligible magnetoresistance signal when the magnetic field is applied, excluding other magnetoresistive effect such as anisotropic magnetoresistance.

Fig. S4. Negligible magnetoresistance in $\text{Fe}_2\text{O}_3/\text{Cr}_2\text{O}_3/\text{Fe}_2\text{O}_3/\text{Ti}$.

Accordingly, we added “**Note that the signal disappears in the junction with Ti as the cap layer, (Supplementary Note 4) which has negligible spin Hall effect, suggesting that the signal is caused by SMR.**” on Page 5, Line 2 in the main-text.

Q2) The authors attribute the orthogonal coupling to a quasi-long range order (QLRO), but I could not find any solid evidence of the QLRO in Cr_2O_3 layer. I think the authors should somehow investigate the spin structure of the Cr_2O_3 layer as a function of the temperature and/or the thickness of Cr_2O_3 layer; otherwise, the proposed scenario is just a speculation. In addition, I cannot understand well why the orthogonal coupling between the Fe_2O_3 is stabilized when the QLRO emerges in Cr_2O_3 . More detailed explanation is desirable.

A: The Reviewer’s comment is reasonable. According to the Reviewer’s suggestions, we theoretically investigate the spin structure of the Cr_2O_3 spacer using the Slonczewski model, where orthogonal interlayer coupling is stabilized by oscillating bilinear coupling and interfacial roughness. When a parallel-preferred area and an antiparallel-preferred area are closed to each other, the Fe_2O_3 cannot form a 180° domain wall to relax the Cr_2O_3 magnetic order in both areas. Hence, there would be a 180° Cr_2O_3 domain wall in the parallel state, and two 90° Cr_2O_3 domain walls in the orthogonal state. The two 90° domain walls can be regarded as a 180° domain wall with twice as much width, which has a smaller energy than the 180° domain wall with original width.

Hence, the orthogonal state has a smaller total energy, leading to the orthogonal coupling.

We rewrote the modelling part on Page 11 and added Fig. 4:

“Non-uniform domain wall state mediated interlayer coupling. Having excluded the magnetic ordering which is uniform in the film plane, we consider magnetic ordering, which is non-uniform in the film plane, as the origin of the interlayer coupling. It is known that orthogonal interlayer coupling could exist in FM/NM/FM trilayers due to the interfacial roughness and oscillating collinear exchange coupling¹³. The collinear interlayer coupling mediated by antiferromagnets also oscillates as a function of the antiferromagnetic layer thickness due to the antiparallel alignment of the magnetic moments of the adjacent monolayers in the antiferromagnet^{7,8}. Hence, the preferred Néel vector orientation of the top and bottom Fe₂O₃ could be either parallel or antiparallel. When a parallel-preferred and an antiparallel-preferred area are close enough to each other, the Fe₂O₃ cannot form a 180° domain wall to relax the Cr₂O₃ magnetic order in both areas. Assuming that the Néel vector in each Fe₂O₃ layer is uniform, the parallel state would induce a 180° domain wall over the Cr₂O₃ thickness t in the antiparallel-preferred area (Fig. 4a). The orthogonal state, however, would induce two 90° domain walls in both areas, which is equal to a 180° domain wall over $2t$ in energy. The 180° domain wall over $2t$ has a lower energy than the 180° domain wall over t , hence the orthogonal state has a lower energy, resulting in the orthogonal interlayer coupling. Taking the further relaxation of the Fe₂O₃ Néel vector and the distance L between the parallel-preferred and the antiparallel-preferred areas into account, the order of the coupling energy can be estimated as $E_c \sim E_{Cr}^2/E_{Fe}$,¹³ where the domain wall energy in Cr₂O₃ is given by

$$E_{Cr} \sim \frac{A_{Cr}}{t} L^2, \quad (1)$$

and the domain wall energy in Fe₂O₃ is given by

$$E_{Fe} \sim \frac{A_{Fe}}{L} L t_t. \quad (2)$$

Here, A_{Cr} and A_{Fe} are the exchange stiffness of Cr₂O₃ and Fe₂O₃, respectively, and t_t is the thickness of the top Fe₂O₃. The resulting coupling energy per area reads

$$E_c \sim q \frac{A_{Cr}^2 L^2}{A_{Fe} t_t^2}, \quad (3)$$

where q is the volume percentage of the in-plane Néel vector that can form this NUDW state. Note that a L^2 factor is subtracted to get the coupling energy per area.

Fig. 4 Schematic of the origin of the orthogonal interlayer coupling. The magnetic order of Cr_2O_3 in parallel-preferred and antiparallel-preferred areas in the collinear state (a) and orthogonal state (b)."

The investigation of spin structure in Cr_2O_3 is very helpful for deepening the understanding of interlayer coupling. But the spin structure in Cr_2O_3 spacer is difficult to investigate experimentally because the detection method such as XMLD is only surface sensitive, which cannot detect the Cr_2O_3 layer in our junction here (The Cr_2O_3 is sandwiched by two Fe_2O_3 layers). To further support our theory model, we plot the maximum coupling field ($\mu_0 H_{\text{MaxCoupling}}$) as a function of the Cr_2O_3 thickness t . The coupling field is inversely proportional to the square of t , which is consistent with our model based on non-uniform domain wall state (Eq. 3 in the main text), verifying the validity of the model and the proposed spin structure in Cr_2O_3 . We added the $\mu_0 H_{\text{MaxCoupling}}-t$ plot to the Supplementary Note 13:

Note 13. Maximum coupling field as a function of Cr_2O_3 thickness

To further investigate the coupling effect and the spin structure in Cr_2O_3 , the maximum coupling field ($\mu_0 H_{\text{MaxCoupling}}$) as a function of Cr_2O_3 thickness is plotted. The maximum coupling field is inversely proportional to the square of the Cr_2O_3 thickness t , (Fig. S14) which is consistent with our model based on the non-uniform domain wall state (Eq. 3 in the main text). Such a consistency verifies the proposed spin structure in Cr_2O_3 in the non-uniform domain wall model.

Fig. S14 Maximum coupling field ($\mu_0 H_{\text{MaxCoupling}}$) as a function of the square of the Cr_2O_3 thickness t . The maximum coupling field is inversely proportional to the square of t .

Accordingly, we added “**The maximum coupling field ($\mu_0 H_{\text{MaxCoupling}}$) is inversely proportional to the square of the Cr_2O_3 thickness t , (Supplementary Note 13) which is consistent with our model based on the non-uniform domain wall (NUDW) state (Eq. 3).**” on Page 12, Line 3 from the bottom in the main text.

Q3) English should be improved. There are many sentences containing grammatical error.

A: We have carefully revise the English writing throughout the whole manuscript.

Reviewers' Comments:

Reviewer #1:

Remarks to the Author:

The authors have adequately responded to my comments, and I recommend publication of their manuscript. Before publication, I suggest to the authors revisit their text to improve the English. Also, I recommend improving the abstract to better articulate the motivation and findings.

Reviewer #3:

Remarks to the Author:

I think the authors reasonably addressed the comments by both reviewers and the revised manuscript is suitable for publication in Nature Communications. However, I think the quality of this manuscript can be further improved by addressing the following two points.

The authors attribute the orthogonal coupling to the co-existence of parallel-preferred and antiparallel-preferred areas due to an interfacial roughness and oscillating collinear exchange coupling, and based on this scenario, they put forward a theoretical model to describe their experimental results. The explanation appears reasonable, but it still remains a matter of speculation since the magnitude of interfacial roughness of their samples is not clear, while a relatively large variation of Cr₂O₃ layer thickness of the order of several angstroms is required for the co-existence of the parallel- and antiparallel-preferred areas according to literatures such as Ref. 7. Thus, I think it is better that the authors present the information of thickness variation and interfacial roughness in their samples.

Secondly, the newly-added Fig. 4 can help the readers to understand their scenario, but this figure can become further helpful by adding a few more words to guide the readers. For example, what is t , where are 90-degree and 180-degree DWs, and where does the magnetic energy increase and why. In addition, the direction of some arrows is not accurate and confusing.

Response Letter of NCOMMS-21-38921B

We very much appreciate the positive evaluations of our manuscript (NCOMMS-21-38921B) by Reviewer #1 (“The authors have adequately responded to my comments, and I recommend publication of their manuscript.”) and Reviewer #3 (“I think the authors reasonably addressed the comments by both reviewers and the revised manuscript is suitable for publication in Nature Communications”). We address the issues raised by them point by point below. And the comments are helpful for our improvements further. Chong Chen are added as co-author for the help on the physical model and English polishing. Amendments of our revised manuscript are summarized below in bold face style.

The main revisions include:

1. We polish the English writing carefully.
2. The roughness data are added.
3. Fig. 4 is modified to be more informative and clear.

Response to Reviewer #1:

The authors have adequately responded to my comments, and I recommend publication of their manuscript. Before publication, I suggest to the authors revisit their text to improve the English. Also, I recommend improving the abstract to better articulate the motivation and findings.

A: We sincerely thank the Reviewer for his/her positive evaluation. We have addressed the Reviewer’s comments and further improved the English writing.

Response to Reviewer #3:

I think the authors reasonably addressed the comments by both reviewers and the revised manuscript is suitable for publication in Nature Communications. However, I think the quality of this manuscript can be further improved by addressing the following two points.

A: We really appreciate the positive evaluations of the Reviewer. We have addressed the Reviewer’s comments as shown below.

Q1) The authors attribute the orthogonal coupling to the co-existence of parallel-preferred and antiparallel-preferred areas due to an interfacial roughness and oscillating collinear exchange coupling, and based on this scenario, they put forward a theoretical model to describe their experimental results. The explanation appears reasonable, but it still remains a matter of speculation since the magnitude of interfacial roughness of their samples is not clear, while a relatively large variation of Cr₂O₃ layer thickness of the order of several angstroms is required for the co-existence of the parallel- and antiparallel-preferred areas according to literatures such as Ref. 7. Thus, I think it is better that the authors present the information of thickness variation and interfacial roughness in their samples.

A: The interfacial roughness of our sample is measured and the corresponding data are added to Supplementary Note 13:

Note 13. Thickness variation and interfacial roughness in Cr₂O₃.

The interfacial roughness in Cr₂O₃ layer is important in our theoretical model. Therefore, the thickness variation and interfacial roughness are measured via atomic force microscope, and the corresponding data are shown in Fig. S14. It can be seen that the variation of Cr₂O₃ thickness is at the order of several angstroms (Fig. S14a). To further support the thickness variation, the roughness data along two diagonal lines are shown in Fig. S14b and c. The thickness variation is large enough (the order of several angstroms) for the co-existence of parallel- and antiparallel-preferred areas in Fe₂O₃ layers^{S13}, further supporting our theory model.

Fig. S14 Thickness variation and interfacial roughness in Cr₂O₃. a, atomic force microscope data. b, c, the roughness data along two diagonal lines.

We modified the sentence on Page 11, Line 4 from the bottom: **“Hence, the preferred Néel vector orientation of the top and bottom Fe₂O₃ can be either parallel or antiparallel because of the thickness variation of Cr₂O₃ layer (Supplementary Note 13).”**

Q2) Secondly, the newly-added Fig. 4 can help the readers to understand their scenario, but this figure can become further helpful by adding a few more words to guide the readers. For example, what is t , where are 90-degree and 180-degree DWs, and where does the magnetic energy increase and why. In addition, the direction of some arrows is not accurate and confusing.

A: Thanks for the Reviewer’s helpful suggestion. As the Reviewer pointed out, more information such as t , 90° and 180° domain walls are added in Fig. 4. To exhibit the enhancement of the magnetic energy more clearly, the equivalent domain walls are added in Fig. 4. 180° domain wall is formed within t in the case of parallel/antiparallel state (Fig. 4a), leading to the energy $E = A_{Cr}/t$. In the case of orthogonal state, the equivalent 180° domain wall is formed within $2t$ (Fig. 4b), resulting in the lower energy $E = A_{Cr}/2t$. Therefore, the orthogonal state is stabilized. In addition, to show the rotation of the direction of arrows more accurately, guidelines for eyes are added.

Fig. 4 Schematic of the origin of the orthogonal interlayer coupling. The magnetic order of Cr_2O_3 in parallel-preferred and antiparallel-preferred areas in the collinear state (a) and orthogonal state (b). The right insets are equivalent magnetic structure in Cr_2O_3 . In collinear state, 180° domain wall is induced over the Cr_2O_3 thickness t . In the orthogonal state, two 90° domain walls equal to a 180° domain wall over $2t$, leading to a lower energy as compared with the collinear state and the stabilization of orthogonal state. The real energy of the two states are more complicated due to the relaxation of the Fe_2O_3 layers. A_{Cr} is the exchange stiffness of Cr_2O_3 . The grey lines are guidelines for magnetic moment rotation.

We modified the sentence on Page 12 Line 4: **“The orthogonal state, however, would induce two 90° domain walls in both areas, which is equal to a 180° domain wall over $2t$ in energy (right inset of Fig. 4b).”**

Reviewers' Comments:

Reviewer #3:

Remarks to the Author:

I think the authors well addressed my previous comment. Now I am pleased to recommend the acceptance of this manuscript for publication as is.

Response to Reviewer #3:

I think the authors well addressed my previous comment. Now I am pleased to recommend the acceptance of this manuscript for publication as is.

A: We really appreciate the positive evaluations of the Reviewer.